# The petrosal and basicranial morphology of *Protoceras celer*

**Selina Viktor Robson** [1]*, **Brendon Seale**[2], **Jessica M. Theodor**[1]

**1** Department of Biological Sciences, University of Calgary, Calgary, Alberta, Canada, **2** Lunenfeld-Tanenbaum Research Institute, Sinai Health System, Toronto, Ontario, Canada

* selina.robson1@ucalgary.ca

## Abstract

Protoceratids are an extinct family of endemic North American artiodactyls. The phylogenetic position of protoceratids in relation to camelids and ruminants has been contentious for over a century. The petrosal morphology of basal (*Leptotragulus*) and derived (*Syndyoceras*) protoceratids has suggested that protoceratids are closely related to ruminants, whereas a prior description of a disarticulated intermediate protoceratid petrosal (*Protoceras celer*) indicated that protoceratids were closely related to camelids. This contradictory evidence implied that there were several character reversals within the protoceratid lineage and brought into question the utility of basicranial characters in artiodactyl phylogenetics. Here, we provide descriptions of an additional *P. celer* petrosal. The descriptions are based on data produced by computed tomography scans, which allowed us to image the petrosal *in situ* in the skull. Our results indicate that the petrosal morphology of *P. celer* is similar to that of other protoceratids, implying that, contrary to previous evidence, petrosal morphology is conserved within the Protoceratidae.

**Data Availability Statement:** Some data cannot be shared publicly because the data are property of the American Museum of Natural History (AMNH). The scans of the Protoceras specimens underlying the results presented in this study are available from MorphoSource.org under the project heading

## Introduction

The Protoceratidae represent an early lineage of North American artiodactyls with elaborate cranial ornamentation. Several of the most basal taxa are hornless, but males of more derived species bear horns on the frontals, parietals, nasals, and/or the occiput [1–3]. Females typically lack horns but bear rough patches in the same locations [2]. Protoceratids range in body mass from 20 kg to 350 kg and are also sexually dimorphic with respect to overall body size [3].

Protoceratids first appeared in the middle Eocene (early Uintan) and persisted into the early Pliocene (latest Hemphillian) of North and Central America [4]. The family is subdivided into the "Leptotragulinae", the Protoceratinae, and the Synthetoceratinae [5]. The "leptotragulines" are a paraphyletic assemblage of basal Eocene hornless forms [4]. The protoceratines consist of most of the smaller horned taxa, including *Protoceras*. Known protoceratine taxa range from the early Oligocene (Whitneyan) to the late Miocene (Clarendonian) [2]. Synthetoceratines first appeared in the early Miocene (early late Arikareean) and persisted until the early Pliocene (late Hemphillian) [6]. The synthetoceratines are larger-bodied, derived protoceratids characterized by their rostral "slingshot" and orbital horns in the males.

"Protoceras celer skulls" (DOIs: https://doi.org/10.17602/M2/M358028; https://doi.org/10.17602/M2/M358040; https://doi.org/10.17602/M2/M358033). These data are copyright of the AMNH and can be downloaded with permission from the AMNH Director of Collections (currently oleary@amnh.org). The comparative specimens used in this study are also available from MorphoSource.org under the project heading "Protoceras celer skulls" (DOIs: https://doi.org/10.17602/M2/M366748; https://doi.org/10.17602/M2/M366743). These data are freely available for download.

**Funding:** This project was funded by an NSERC Discovery Grant (https://www.nserc-crsng.gc.ca/professors-professeurs/grants-subs/dgigp-psigp_eng.asp) awarded to JMT. The funders had no role in study design, data collection and analysis, decision to publish, or preparation of the manuscript

**Competing interests:** The authors have declared that no competing interests exist.

Apart from the presence of cranial appendages, protoceratids exhibit a morphology typical of generalized selenodont artiodactyls, including a basic selenodont dentition. Protoceratids have elongated limbs and a fused ectomesocuneiform, but their cuboid and navicular remain separate and their metapodial keels are incomplete [4]. Protoceratines and synthetoceratines have a complete postorbital bar, but this condition is not present in basal members of the family [7].

The phylogenetic affinities of protoceratids have been the subject of considerable dispute. Protoceratids were originally allied with ruminants, a view that persisted for half a century [8–16]. Like most ruminants, derived protoceratids lack upper incisors and possess an incisiform lower canine. The protoceratid auditory bulla is hollow and is compressed between the glenoid fossa and the exoccipital. Yet protoceratids lack a cubonavicular, one of the most distinctive ruminant synapomorphies [17].

"Leptotragulines" have historically been placed in Tylopoda [11–13], but the more derived protoceratids were not allied with camelids (and other tylopods) until the mid-twentieth century [2, 6, 18–24]. This shift in systematics was largely driven by morphological similarities between protoceratids and camelids. It is now understood that most of these similarities are plesiomorphic (e.g., incomplete metapodial keels, unfused cuboid and navicular) or homoplastic (e.g., elongate limbs, complete postorbital bar) [12, 16, 18, 20, 23, 25]. The one unusual morphology shared by protoceratids and camelids is the location of the vertebrarterial canal—both families have a vertebral artery canal that passes through the pedicles of the cervical vertebrae. This condition is only found in camelids, protoceratids, and the endemic European xiphodontids [4, 21]. However, protoceratids lack other morphologies that have been associated with camelids, such as the presence of a dorsally-projecting angular hook on the dentary and an inflated auditory bulla filled with cancellous bone [4].

This conflicting osteological evidence has presented challenges for inferring protoceratid relationships. At the turn of the twenty-first century, novel information became available. The endocranial morphology of the basal "leptotraguline" protoceratid *Leptotragulus* was described from physical dissections of fossils [7] and the derived synthetocerine protoceratid *Syndyoceras* was described from computed tomography (CT) scans [25]. Based on these descriptions, Joeckel and Stavas [25] and Norris [7] concluded that protoceratid endocranial morphology is more similar to that of ruminants than to that of camelids, suggesting that early workers may have been correct in placing protoceratids with ruminants [7, 25].

An additional description of a protoceratid petrosal was provided by O'Leary [26]. This detailed description was of AMNH-VP 645, a skull and disarticulated petrosal attributed to *Protoceras celer* [26]. This specimen, in contrast to the UNSM 1153 *Syndyoceras* material and the YPM and MCZ *Leptotragulus* material described by Joeckel and Stavas [25] and Norris [7], showed a deep subarcuate fossa. The petrosal characters for *P. celer* were coded in a phylogenetic analysis based on AMNH-VP 645 [27]. The total evidence phylogenetic analysis recovered protoceratids in a position within Ruminantia, but the morphological phylogenetic analysis recovered protoceratids in a position close to camelids, supporting the interpretation that protoceratids are tylopods [27].

The description of AMNH-VP 645 calls into question characters for *Syndyoceras* [25] and differs from the description of *Leptotragulus* [7]. There are two potential explanations for these discrepancies: *P. celer* represents several character state reversals within Protoceratidae, or the AMNH-VP 645 petrosal is incorrectly attributed to *P. celer*. We tested these two scenarios by subjecting two skulls of *P. celer* [AMNH-VP 1229; AMNH-VP 53523] to CT scanning and reconstructed the petrosal from the CT scan data. With these additional data, we were able to compare the petrosal morphology of AMNH-VP 53523 to that of AMNH-VP 645.

## Materials and methods

### Institutional abbreviations

AMNH-VP, American Museum of Natural History, New York; UCMZ, University of Calgary Museum of Zoology, University of Calgary; MCZ, Museum of Comparative Zoology, Harvard University; UNSM, University of Nebraska State Museum paleontology collections, University of Nebraska, Lincoln; YPM, Yale Peabody Museum, Yale University; ZM, University of Nebraska State Museum mammalogy collections.

### Material

AMNH-VP 1229 and AMNH-VP 53523 are skulls, referred to *Protoceras celer*, from the Pole-slide member of the Brule Formation, South Dakota. Both specimens are of Whitneyan age (early Oligocene) [2], approximately 31.4 to 30.0 million years old [28].

AMNH-VP 1229 and AMNH-VP 53523 were both referred to *P. celer* by Patton and Taylor [2] on the basis of dental and cranial features. Compared to other protoceratine protoceratids, the orbits of *Protoceras* are more rostral and the facial region is longer. The cranium of *Protoceras* is shorter than that of other protoceratines, but a pronounced sagittal crest is retained. An occipital horn is absent. *Protoceras* males have cranial appendages on the maxilla, above the orbit, and on the parietal. The P1 of *Protoceras* is equidistant between the canine and the P2. The P2 and P3 are anteroposteriorly elongate, and the P3 has a strongly developed protocone. Compared to other protoceratines, the upper molars have a more pronounced lingual cingulum and are lower crowned.

The right side of AMNH-VP 1229 has minor dorsoventral compression, but the specimen is mostly complete. There is slight damage to the dorsal skull roof, and the ventral portion of the left orbit is missing. AMNH-VP 1229 is identified as a female because it lacks the cranial ornamentation present in males and is smaller in size (Fig 1A–1C).

AMNH-VP 53523 has not been completely prepared and matrix remains on much of the basicranium. The skull is crushed dorsoventrally but maintains its original width. Cranial appendages are present but damaged, aside from the intact right rostral horn. AMNH-VP 53523 is identified as a male because of the presence of cranial appendages and larger size (Fig 1D–1F).

### Computed tomography scans

AMNH-VP 1229 and AMNH-VP 53523 were subjected to micro-computed tomography (μCT) scanning at the High-Resolution Computed Tomography Facility at the University of Texas at Austin. Both skulls were initially scanned at a voxel size of 50 μm using the P250D x-ray detector operating at 419 kV and 1.8 μA. These scans produced a stack of 140 images for AMNH-VP 1229 and a stack of 151 images for AMNH-VP 53523. AMNH-VP 1229 was found to have several high-density deposits in the basicranial region. These high-density deposits distorted the CT images and removed AMNH-VP 1229 as a candidate for high-resolution imaging.

The basicranium of AMNH-VP 53523 was subsequently scanned at voxel dimensions of 63.4765 x 63.765 x 0.07436 μm using the II x-ray detector operating at 210 kV and 0.11 μA. This produced a set of 300 slices, covering approximately 22.308 mm of the basicranium, starting at the occipital condyles and ending just rostral to the petrosal.

Cranial morphologies were reconstructed from the CT scans using Amira 5.3 for Mac OS X (Visage, Inc., Chelmsford, MA: http://www.visage.com).

The scans of the *Protoceras* specimens underlying the results presented in this study are available from MorphoSource.org under the project heading "*Protoceras celer* skulls" (DOIs:

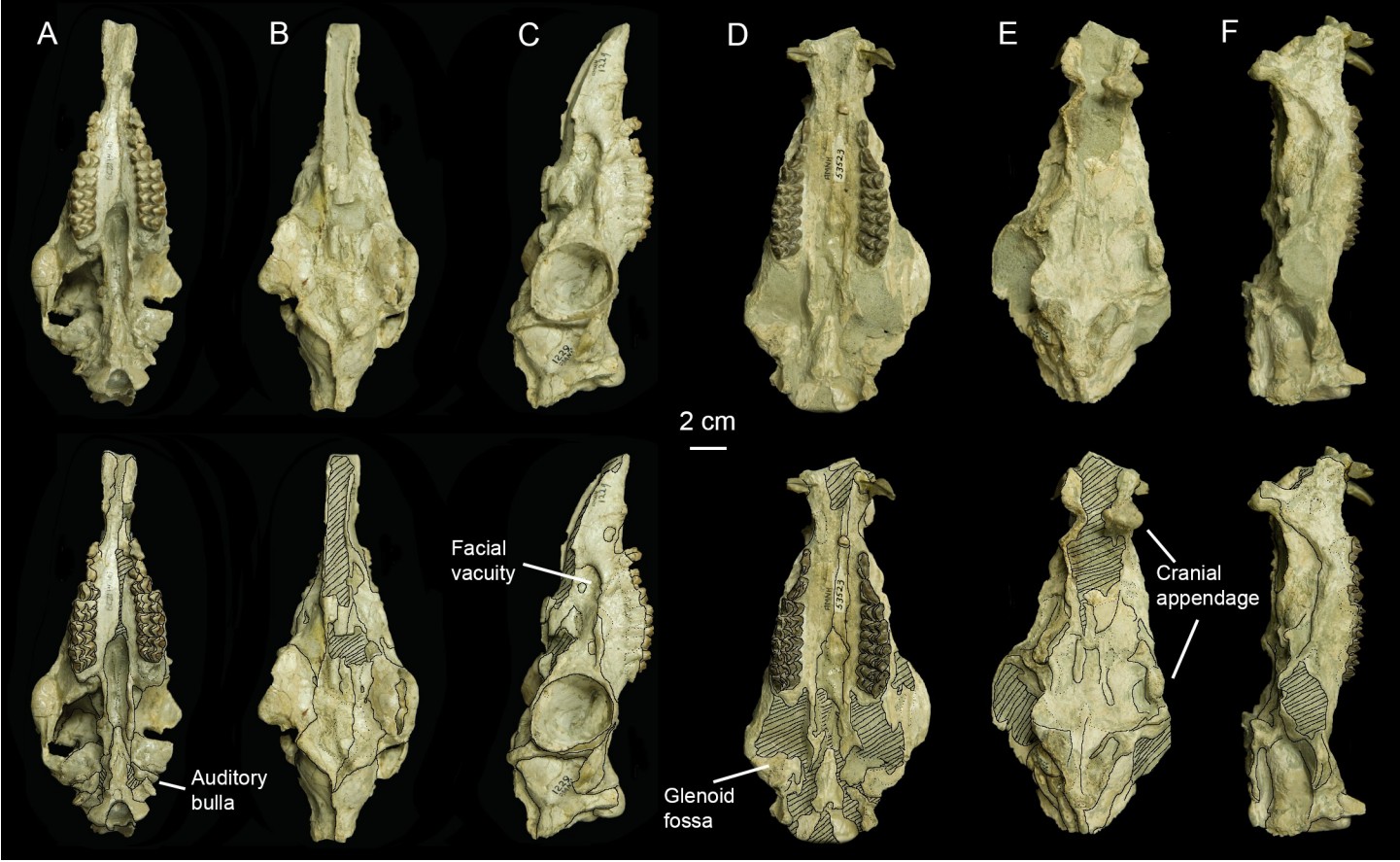

**Fig 1. Photographs of the *Protoceras celer* specimens included in this study.** (A) Ventral view of AMNH 1229. (B) Dorsal view of AMNH 1229. (C) Right lateral view of AMNH 1229. (D) Ventral view of AMNH 53523. (E) Dorsal view of AMNH 53523. (F) Right lateral view of AMNH 53523.

https://doi.org/10.17602/M2/M358028; https://doi.org/10.17602/M2/M358040; https://doi.org/10.17602/M2/M358033). These data are copyright of the AMNH and can be downloaded with permission from the AMNH Director of Collections (currently oleary@amnh.org).

Comparative specimens (UCZM 1975.496; UCMZ 1989.47) were CT scanned at the University of Calgary. The *Camelus dromedarius* (UCMZ 1975.496) specimen was scanned at the Centre for Mobility and Joint Health, McCaig Institute for Bone and Joint Health, University of Calgary, using a Dual-energy CT/GSI (GE Revolution HD GSI, 140 kV and 80 kV fast switching), at a resolution of 527.433 x 527.344 x 265.000 μm voxels The *Muntiacus* (UCMZ 1989.47) specimen was scanned at the University of Calgary Micro-CT Laboratory using a Sky-Scan1173 operating at 80 kV and 60 μA, producing a scan with voxel dimensions of 71.00 x 71.00 x 71.00 μm. The comparative specimens used in this study are also available from MorphoSource.org under the project heading "*Protoceras celer* skulls" (DOIs: https://doi.org/10.17602/M2/M366748; https://doi.org/10.17602/M2/M366743). These data are freely available for download.

## Measurements

All measurements were taken using the 3D measurement tool of Amira. Basicranial length measurements were based on the protocols outlined by Janis [29]. Total skull lengths were measured from the tip of the rostrum to the caudal-most point of the occiput. Length and

width measurements of the anterior semicircular canal were made following the protocol of Spoor et al. [30], and the arc radius was calculated using the equation provided by Ekdale [31]. Height and width measurements of the cochlea were made following Ekdale [32].

## Body mass estimates

Body mass (BM) estimates were calculated for AMNH-VP 1229 and AMNH-VP 53523. Estimates were based on the predictive body mass regressions proposed by Janis [29]. We used both the "all ungulates" and the "ruminants only" total skull length (SL) and basicranial length (BL) regressions to estimate body mass. We chose to use the "ruminants only" regressions because, despite the phylogenetic position of protoceratids being uncertain, Janis [29] the cranial morphology of *P. celer* greatly resembles that of a ruminant.

The two "all ungulates" body mass equations used are:

$$Total\ skull\ length: \ \log_{10} BM\ (kg) = 2.975(\log_{10} SL) - 2.344$$

$$Basicranial\ length: \ \log_{10} BM\ (kg) = 3.137(\log_{10} BL) - 1.062$$

The two ruminant body mass equations used are:

$$Total\ skull\ length: \ \log_{10} BM\ (kg) = 2.969(\log_{10} SL) - 2.348$$

$$Basicranial\ length: \ \log_{10} BM\ (kg) = 3.281(\log_{10} BL) - 1.209$$

## Agility scores

Agility scores (AGIL) for AMNH-VP 53523 were calculated using the anterior semicircular canal radius (ASCR) "all mammals" predictive equation of Silcox et al. [33]. This is because only the anterior semicircular canal was preserved in enough to detail to measure the width and height. We used two body mass estimates, based on different cranial variables, in our calculations. This provided a range of likely agility scores. The anterior semicircular canal equation is:

$$ASCR: \ \log_{10} AGIL = 0.850 - 0.153(\log_{10} BM) + 0.706(\log_{10} ASCR)$$

Body mass in the AGIL predictive equation is in grams, whereas the body masses calculated from the Janis [29] regressions are in kilograms. As such, a simple conversion is required.

## Results

The external cranial morphology of *Protoceras* was thoroughly described by previous authors [2, 8–10, 18] so only a brief description of external morphology will be presented here. AMNH-VP 1229 is better preserved externally and AMNH-VP 53523 is better preserved internally. As such, descriptions are based on a composite of the two skulls, with external descriptions primarily based on AMNH-VP 1229 and endocranial descriptions primarily based on AMNH-VP 53523.

## Rostrum, orbit, and cranial vault

The preorbital region is long and narrow, comprising approximately 2/3 of the total skull length (Fig 1). The nasal bones are small and the external nares are large, spanning the majority of the rostrum. The nasals meet at a pointed process above the external nares. AMNH-VP

53523 has rostral horn-like cranial appendages on the maxillaries, caudal to the nasals (Fig 1E and 1F).

There are facial vacuities on the rostrum at the level of P3 (Fig 1C and 1F). These vacuities have a well-defined rostral margin and an indistinct caudal margin. On AMNH-VP 1229, the palatine canal opens as a small foramen on the ventrocaudal edge of the left facial vacuity. A crest extends from the ventrocaudal margin of the vacuity to the anterior margin of the orbit. The dorsal surface of this crest is textured. AMNH-VP 1229 has a distinct infraorbital foramen just rostral to the orbit (Fig 1C).

The orbits are large with a complete postorbital bar. On AMNH-VP 53523, there are cranial appendages projecting upwards from the dorsal border of the orbits (Fig 1E). The orbital bones are thin, and the sutures are difficult to distinguish. The lacrimal appears to be a large bone pierced ventrally by the lacrimal canal. The zygomatic arch slopes ventrally from the squamosal to the orbit (Fig 1C). The interorbital area (comprising the frontals) is mostly flat with a slight caudal incline (Fig 1C and 1F). Two distinct, bilateral crests originate from the interorbital region, one directed rostrally and the other directed caudally. The rostral crests extend anteriorly onto the nasals. The caudal crests originate at the dorsocaudal margin of the orbit and extend posteriorly as bilateral sagittal crests, eventually joining in the midline of the occiput and then intersecting with the shield-like nuchal crest. On AMNH-VP 53523, the sagittal crests become the parietal cranial appendages (Fig 1E). The parietals are smooth with no distinctive foramina or projections, except for a short zygomatic process that contributes to the postorbital bar.

The dentition of *P. celer* is fully described in previous publications [2, 18]. Both skulls have canines; however, the canines of AMNH-VP 1229 are greatly reduced compared to those of AMNH-VP 53523 (Fig 1A and 1D). The palate is narrow and flat. The palatine crests and the pterygoid processes of the sphenoid are tall, and the internal nares are visible along the midline. The palatal region is mediolaterally constricted.

## Petrosal

Most of the petrosal was captured in the high-resolution CT scan of AMNH-VP 53523 (Fig 2). The caudal portion of the mastoid region (along with other caudal structures) was not included, but the morphology of the petrosal can still be described.

The promontorium is hemi-ellipsoid with a well-rounded lateral face (Fig 2A). A small epitympanic wing, which lacks a lateral process, projects rostrally from the anterior margin of the promontorium (Fig 2A and 2C). The epitympanic wing is roughly triangular and forms the rostral-most part of the petrosal. A groove separates the epitympanic wing from the posteromedial flange, which begins just caudal to the epitympanic wing and projects ventrally from the lower margin of the promontorium (Fig 2A). The rostral tympanic process is absent.

The promontorium lacks a transpromontorial sulcus and a stapedial artery sulcus. A circular, ventrocaudally directed fenestra cochleae opens at the caudal end of the promontorium (Fig 2A and 2D). There is an indistinct caudal tympanic process posterior to the fenestra cochleae. The fenestra vestibuli is an oval opening dorsal to the fenestra cochleae, and a small secondary facial foramen lies just dorsal to the fenestra vestibuli (Figs 2A, 2D and 3C). The path of the facial canal can be briefly traced internally from the secondary facial foramen, but quickly disappears.

A deep and circular fossa for the muscularis tensor tympani excavates the tegmen tympani just rostral to the fenestra vestibuli and the secondary facial foramen. The stapedial muscle fossa is a deep and wide depression directly caudal to the fenestra vestibuli and the secondary facial foramen (Fig 2A). The stapedial muscle fossa terminates ventrally as the stylomastoid

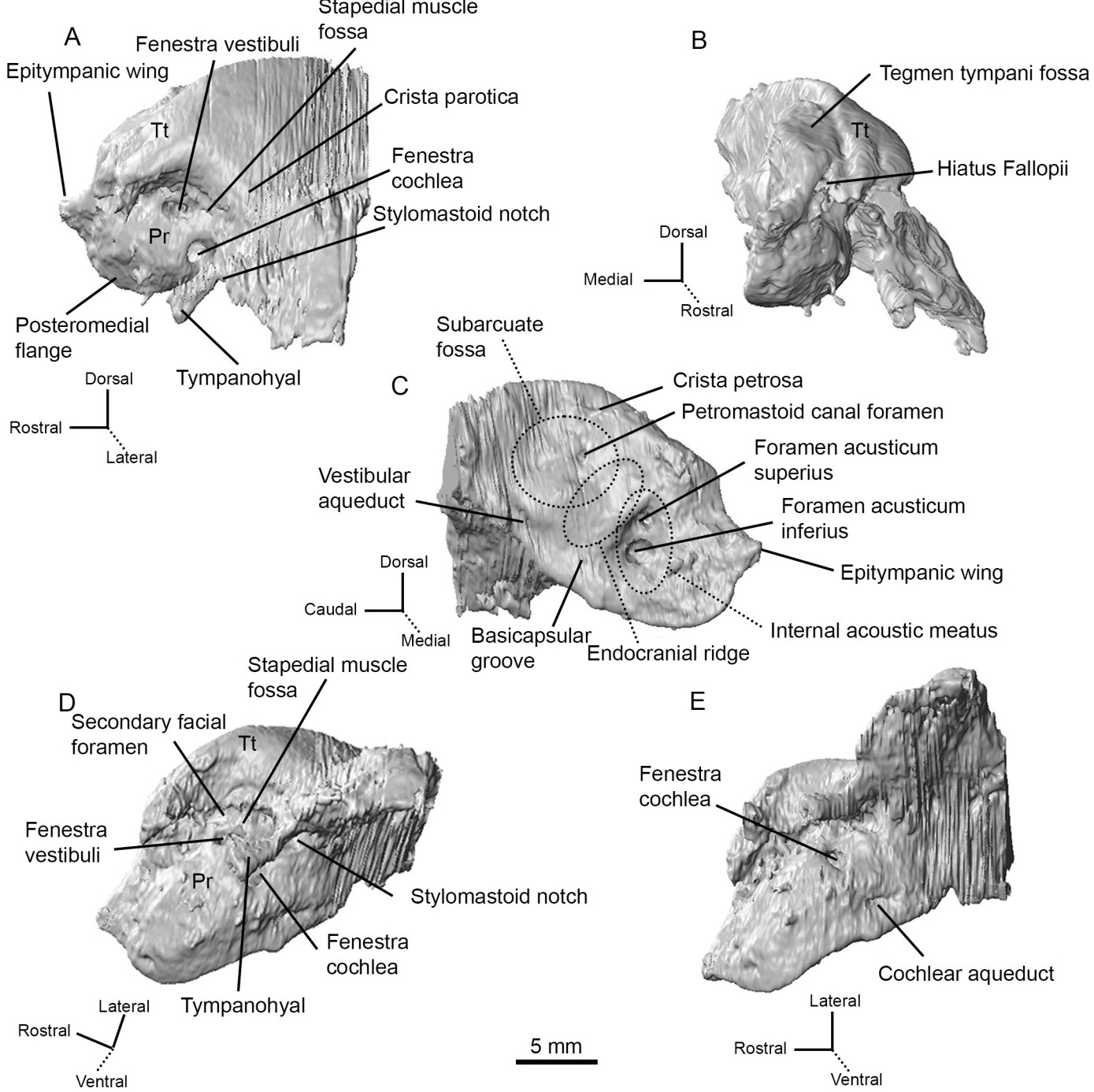

**Fig 2. CT renderings of the left petrosal of AMNH 53323 in five orientations.** (A) Lateral (tympanic) view. (B) Rostral view. (C) Medial (endocranial) view. (D) Ventrolateral view. (E) Ventral view. Abbreviations: Pr, promontorium; Tt, tegmen tympani.

notch, which is the petrosal contribution to the stylomastoid foramen (Fig 2A and 2D). The rest of the stylomastoid foramen is formed by the exoccipital and represents the exit of the facial nerve from the middle ear cavity.

On the pars canalicularis, the tegmen tympani is moderately inflated with a distinctive, oval-shaped tegmen tympani fossa on the dorsomedial side (Fig 2B). The tegmen tympani is

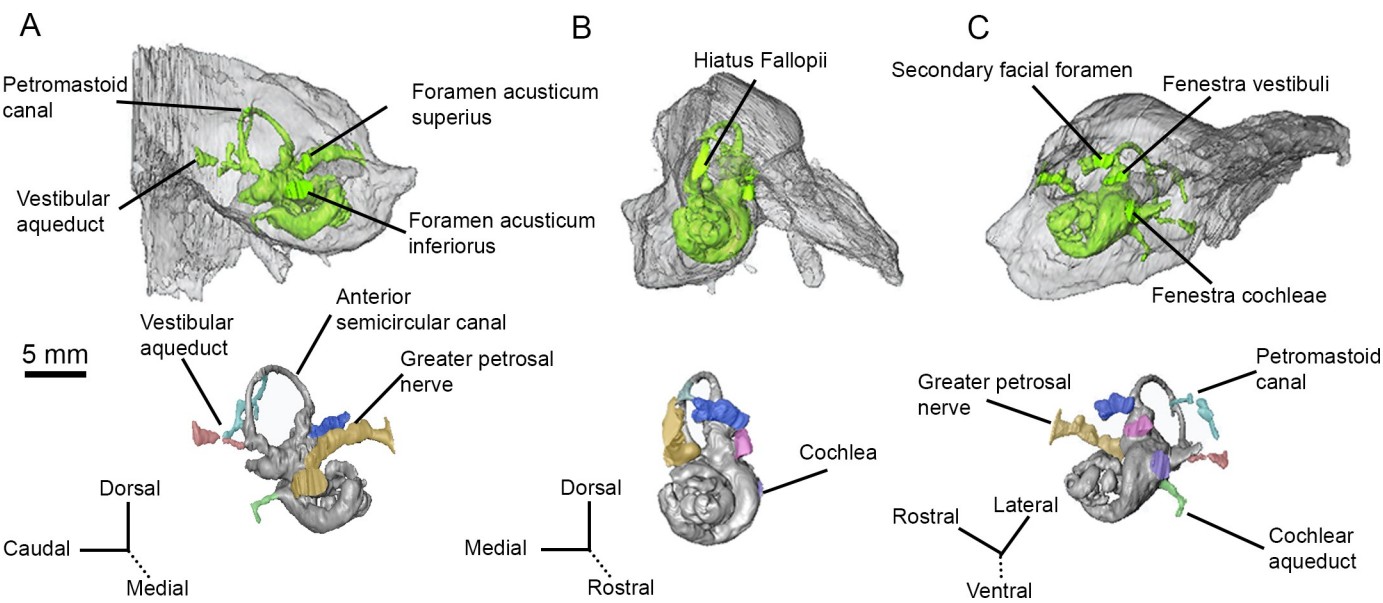

**Fig 3. CT renderings of the bony labyrinth (and surrounding petrosal, upper images) of AMNH 53323.** (A) Medial (endocranial) view. (B) Rostral view. (C) Ventrolateral view.

pierced rostrally by a slit-like hiatus Fallopii (Figs 2B and 3B). The path of the greater petrosal nerve can be traced from where it enters the foramen acusticum superius with the rest of the facial nerve to where it exits though the hiatus Fallopii (Fig 3). The exact point at which the greater petrosal nerve diverges from the rest of the facial nerve cannot be located because the facial canal is incomplete. The greater petrosal nerve canal is slightly exposed at the rostral end of the epitympanic recess, inside the fossa muscularis tensor tympani, just ventrolateral to where the nerve emerges through the hiatus Fallopii. This exposure may be the result of thin bone that has been eroded.

The lateral portion of the tegmen tympani curves ventrally to form the roof of the epitympanic recess, which is an elongated channel that originates caudal to the epitympanic wing and terminates at the stapedial muscle fossa (Fig 2A and 2D). The epitympanic recess lacks a distinct fossa for the head of the malleus. A short crista parotica, situated caudal to the stapedial muscle fossa, separates the epitympanic recess from the mastoid region of the petrosal (Fig 2A). The tympanohyal projects laterally from the crista parotica (Fig 2A and 2D). The lateral border of the tympanohyal is indistinct and may either be broken or merged with the ectotympanic.

The mastoid region comprises more than half of the petrosal. The caudal part of the mastoid region was not captured in the high-resolution CT scan of AMNH-VP 53523, but the mastoid region is clearly large and wedge shaped (Fig 2). There is a distinct mastoid process projecting ventrolaterally from the caudal portion of the mastoid region (Fig 2). As has been described previously [10], this mastoid process is exposed externally as a strip of bone sandwiched between the exoccipital and the squamosal (Fig 4B). A mastoid plate (see O'Leary [26]) is not present.

The tegmen tympani forms a right angle with the endocranial surface of the petrosal, and a short crista petrosa rostral to the subarcuate fossa separates the tegmen tympani fossa (see Orliac and O'Leary [34]) from the endocranial face (Fig 2C). The internal acoustic meatus is deep with a smooth border. The foramen acusticum superius and foramen acusticum inferius are separated by a narrow crista transversa (Figs 2C and 3A). The foramen acusticum inferius

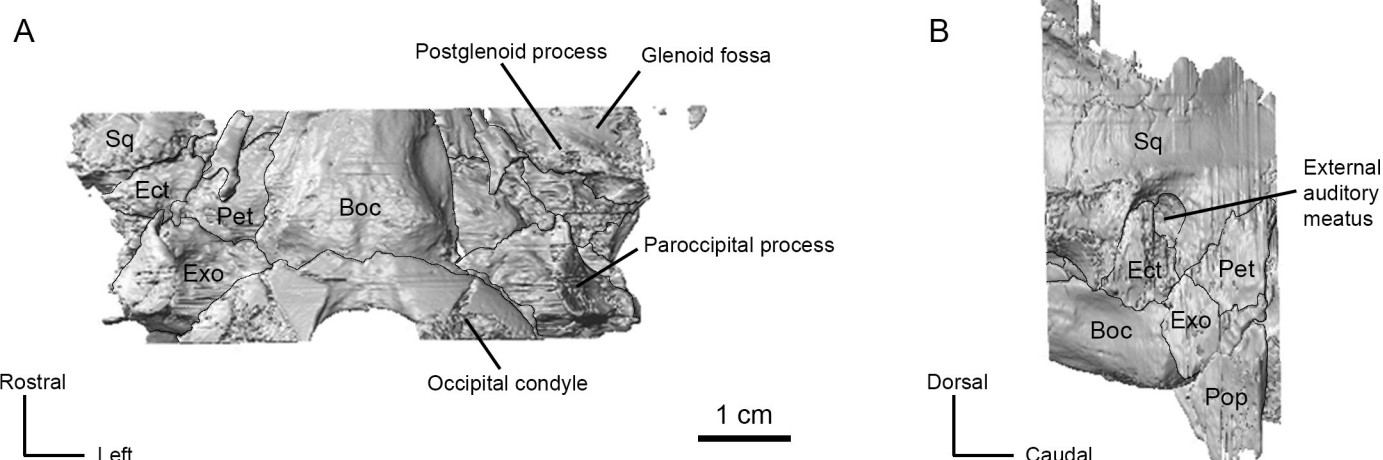

**Fig 4. CT renderings of the basicranium of AMNH 53323.** (A) Ventral view. (B) Left lateral view. Abbreviations: Boc, basioccipital; Ect, ectotympanic; Exo; exoccipital; Pop; paroccipital process of exoccipital; Sq, squamosal.

is large and opens caudally whereas the foramen acusticum superius is small and opens ventrally. A prefacial commissure borders the dorsal side of the internal acoustic meatus, but no prefacial commissure fossa is present. The subarcuate fossa lies caudal to the internal acoustic meatus. The subarcuate fossa is wide and extremely shallow, appearing as a subtle depression in the petrosal. A petromastoid canal is present on the rostral border of the subarcuate fossa (Figs 2C and 3). Internally, the petromastoid canal passes just inside the arc of the anterior semicircular canal, terminating halfway between the endocranial face and tympanic face of the petrosal.

The vestibular aqueduct, which carried the endolymphatic duct, travels from the common crus of the semicircular canals to emerge on the endocranial surface of the petrosal, ventrocaudal to the subarcuate fossa (Figs 2C and 3). A basicapsular groove (= petrobasilar canal [7]) runs along the ventral border of the petrosal (Fig 2C). The cochlear aqueduct, on the ventromedial surface of the petrosal, sits medial to the basicapsular groove and slightly caudal to the internal acoustic meatus (Figs 2E and 3). Internally, the cochlear aqueduct originates just medial to the fenestra cochleae and is directed posteriorly as a long, thin channel.

## Bony labyrinth

Sections of both the left and right bony labyrinths are preserved in AMNH-VP 53523. The left bony labyrinth is more complete and will be the basis of this description (Fig 3). The cochlear canal makes approximately 2.75 turns (rotation of 990˚), but the exact termination point of the apex cannot be identified. Several sections of the cochlear canal are infilled with sediment, obscuring the borders and making it unclear whether the basal and secondary turns naturally contact each other. The aspect ratio, calculated by dividing the height of the spiral by the width of the basilar turn [32], is approximately 0.80.

The vestibule is represented by a slightly bulbous saccule (spherical recess) and utricle (elliptical recess). The saccule, which is a medial bulge extending from the fenestra vestibuli, is more inflated than the utricle. The utricle sits between the saccule and the anterior ampulla of the anterior semicircular canal. The anterior semicircular canal is the only semicircular canal fully preserved in the left bony labyrinth (Fig 3). The posterolateral base of the lateral semicircular canal is present, but the path of the canal cannot be traced. No part of the posterior semicircular canal could be reliably identified; a structure identified as the medial portion of the

vestibular aqueduct may include the root of the posterior semicircular canal, but this cannot be confirmed. Fragments of both the anterior and posterior semicircular canals, including the common crus, are present in the right bony labyrinth. The right lateral semicircular canal could not be located.

The left anterior semicircular canal is sigmoidal and lies in more than one plane. The anterior portion of canal projects rostrally, throwing that part of the semicircular canal into a tight arc. The path of the canal is less curved posteriorly, becoming almost straight in the region of the common crus.

Other aspects of the bony labyrinth are discussed along with the morphology of the petrosal.

## Ectotympanic

The lateral portion of the ectotympanic is present in AMNH-VP 53523. The ectotympanic comprises the entirety of the *Protoceras* auditory bulla [10], but the bullar portion of the bone is missing from the specimen. AMNH-VP 1229 has a superficially complete auditory bulla but the internal structures are not preserved (Fig 1A). The bulla is small and uninflated and the anteromedial side projects as a wide and blunt styliform process. The bullar portion of the ectotympanic sits between the squamosal, basioccipital, and paroccipital process of the exoccipital. There is a gap between the bulla and the basioccipital in AMNH-VP 1229, but no internal structures, including the petrosal, can be seen because of poor internal preservation.

The external auditory meatus is located between the postglenoid process and post-tympanic process of the squamosal (Fig 4B). Both the squamosal and the ectotympanic contribute to the external auditory meatus; the rostral and ventral borders of the meatus are formed by the dorsal margin of the ectotympanic, and the dorsal and caudal borders of the meatus are formed by the squamosal (Fig 4B). There is a gap between the postglenoid process and the rostral face of the ectotympanic, but the caudal face of the ectotympanic and the post-tympanic process are in articulation. The ectotympanic extends as a compressed plate ventral to the external auditory meatus. The ventral border of this plate is missing in both specimens, but CT scans of AMNH-VP 53523 show that the plate is filled with cancellous bone.

## Squamosal

The glenoid fossa of the squamosal is mediolaterally elongate with a slightly convex articular surface (Figs 1A, 1C and 4A). A small, non-pneumatized postglenoid process borders the glenoid fossa. The postglenoid foramen penetrates the caudal face of the postglenoid process. Internally, contact between the squamosal and the petrosal is interrupted by a sinus venosus temporalis (Fig 5C). The presence of a foramen jugular spurium, an opening for the sinus venosus temporalis, cannot be confirmed because the bony elements are not in tight articulation. The presence of a glenoid foramen cannot be confirmed for the same reason.

A large rostrocaudally directed canal runs through the ventral part of the squamosal, piercing the skull above the glenoid fossa. We identify this exit as the supraglenoid foramen based on AMNH-VP 1229. A similar foramen could not be identified on the surface of AMNH-VP 53523, but the internal canal is clearly visible in CT cross-sections (Fig 3A). The canal appears to terminate caudally around the rostral margin of the ectotympanic, but the exact point of termination is indistinct.

## Exoccipital

The exoccipital of *P. celer* is dominated by a prominent paroccipital process that projects ventrolaterally, extending well beyond the ventral margin of the basioccipital (Fig 4). A crest on

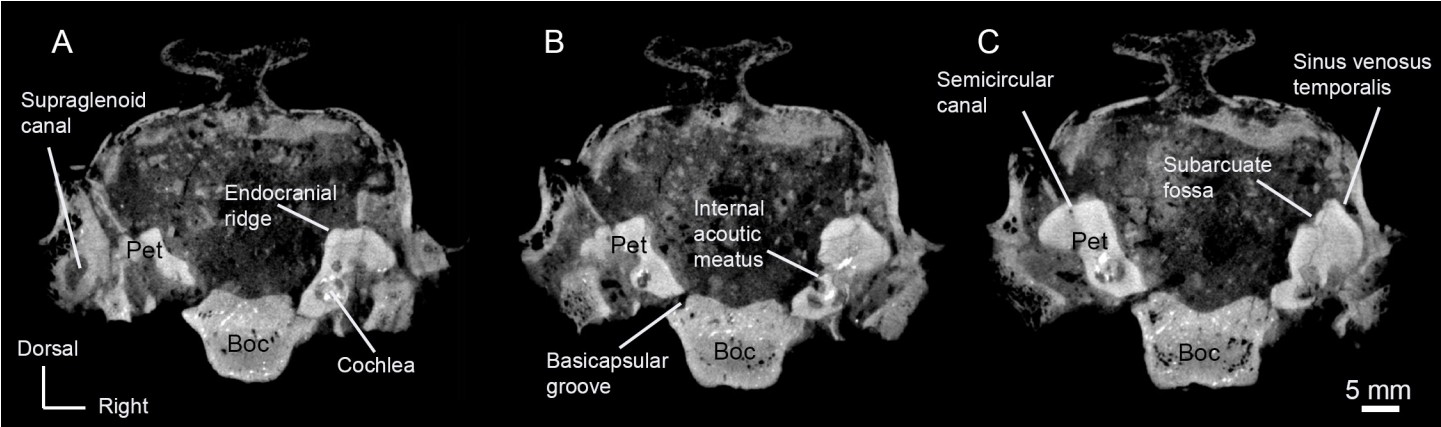

**Fig 5. Transverse CT slices of AMNH 53323 showing important morphological features.** (A) Slice 88. (B) Slice 107. (C) Slice 131. Abbreviations: Boc, basioccipital; Pet; petrosal.

the lateral side of the paroccipital process intersects with the nuchal crest. The mastoid portion of the petrosal is visible laterally as a narrow strip of bone between the ventral margin of the squamosal and the paraoccipital process. Based on AMNH-VP 1229, the paroccipital process and the ectotympanic bulla are in close contact (Fig 1A).

## Basisphenoid

The exact point of contact between the basioccipital and basisphenoid is ambiguous because of a transverse crack through the region on AMNH-VP 53523 (Fig 1D). The basisphenoid is broad caudally and narrow rostrally, forming a rod that is bordered laterally by the pterygoid processes of the alisphenoid (Fig 1A and 1D). The ventral surface of the basisphenoid has two longitudinal grooves, one on each side of the midline. The foramen ovale is externally visible on the left lateral side of AMNH-VP 1229, ventral to the otic region.

## Basioccipital

The basioccipital is bounded dorsolaterally by the exoccipitals and rostrally by the basisphenoid. The basioccipital and exoccipitals are tightly sutured. The basioccipital is a robust bone with a groove running along the ventral midline (Fig 4A). The large occipital condyles extend from the exoccipital onto the basioccipital with paired tubercles at their anteroventral margin (Fig 4A). The dorsolateral border of the condyle is demarcated by a distinct groove, and the hypoglossal foramen is located on the dorsal aspect of this groove. The left side of both AMNH-VP 1229 and AMNH-VP 53523 has two adjacent foramina in this position, likely a separate hypoglossal foramen and condylar foramen.

A paired groove is present on the dorsolateral (endocranial) surface of the basioccipital where the basioccipital is close to contacting the ventral margin of the petrosal (Fig 5B). This groove is interpreted as the basicapsular groove, which carries the inferior petrosal venous sinus. The groove is only present on the basioccipital for a small section, suggesting that the path of the sinus diverges from the bone rostrally.

## Body masses and agility scores

Body mass and agility scores were calculated for AMNH-VP 53523. The rostral to caudal skull length of AMNH-VP 535253 is 18.8 cm, and the basicranial length is 6.21 cm. We also

calculated the body mass of AMNH-VP 1229 for comparative purposes. The rostral to caudal skull length of AMNH-VP 1229 is 20.75 cm, and the basicranial length is 4.78 cm. The total skull length of AMNH-VP 53523 is less than that of AMNH-VP 1229 because the anterior-most part of the rostrum of AMNH-VP 53523 is missing. We estimated body masses using the total skull length of both specimens, but we did not use these estimates when predicting agility scores for AMNH-VP 53523; we only calculated agility scores using the body masses predicted from the basicranial length.

Using the "all ungulates" regressions, the body mass of AMHN-VP 53523 was estimated to be 28.06 kg (based on total skull length) and 26.60 kg (based on basicranial length). Using the "ruminants only" regressions, skull measurements of AMNH-VP 53523 provided body mass estimates of 27.32 kg (based on total skull length) and 24.66 kg (based on basicranial length).

Total skull length and basicranial length provided body mass estimated of 37.51 kg and 11.71 kg for AMNH-VP 1229, respectively, when used in the "all ungulates" regressions. The "ruminants only" regressions produced body mass estimates of 36.50 kg (based on total skull length) and 10.45 kg (based on basicranial length). The width of the anterior semicircular canal of AMNH-VP 53523 is 5.48 mm and the height of the anterior semicircular canal is 5.15 mm—the arc radius is 2.66 mm. When applied to the appropriate agility predictive equation (see Materials and Methods), we recover two agility scores. Using the body mass based on the "all ungulates" regression, we predict an agility score of 3.29. Using the body mass based on the basicranial length, we predict an agility score of 3.00.

## Discussion

### Petrosal

The *P. celer* petrosal is typical of protoceratids. It lacks the ventromedial flange characteristic of both basal and extant camelids, the homacodontid *Bunomeryx*, *Merycoidodon culbertsoni*, and *Cainotherium* (see 'Discussion: Basioccipital' for further discussion), [7, 25, 35–37], and there is an endocranial ridge separating the cerebral and cerebellar faces (Fig 5A), a feature shared with other protoceratids, with ruminants, and with anoplotheriids (Fig 6) [7, 25, 38, 39]. The presence of this ridge in *P. celer* indicates that a clear cerebral/cerebellar division was maintained throughout protoceratid evolution. This morphology has been used as evidence that protoceratids should be allied with ruminants [7, 25], but the distribution of this morphology is not well-documented in other artiodactyl groups.

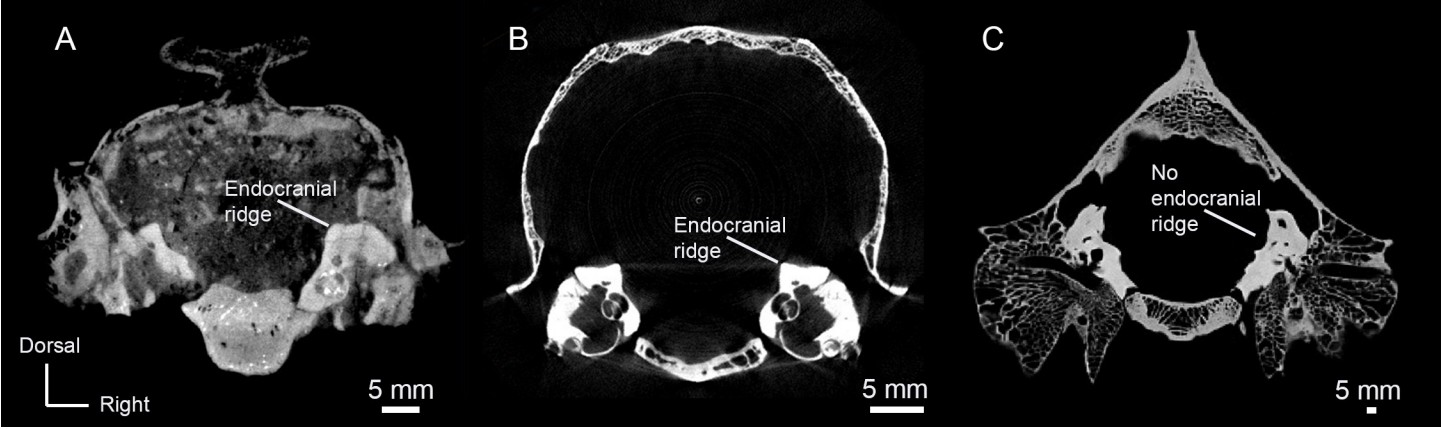

**Fig 6. Transverse CT slices of *Protoceras*, a ruminant, and camelid showing differences in the endocranial ridge.** (A) Slice 88 of *Protoceras celer*, AMNH-VP 53523. (B) Slice 633 of *Muntiacus* (ruminant), UCMZ 1989.47. (C) Slice 338 of *Camelus dromedarius* (camelid), UCZM 1975.496.

Like other protoceratids, the subarcuate fossa of *P. celer* is a shallow depression on the endocranial face, and there is no mastoid fossa. The subarcuate fossa houses the paraflocculus of the cerebellum in life [40]. The depth of the subarcuate fossa varies among artiodactyls, and the shallow nature of the protoceratid subarcuate fossa has been used as an argument for uniting protoceratids with pecoran ruminants [7, 25]. This is because pecoran ruminants also have a shallow subarcuate fossa, whereas the basal camelid *Poebrotherium* and extant camelid *Lama glama* have a deep subarcuate fossa [25, 26, 41]. Camelids are not the only artiodactyls with a deep subarcuate fossa: the early artiodactyls *Bunomeryx*, *Diacodexis ilicis*, *Dichobune*, and *Gobiohyus* also have a deep subarcuate fossa [34, 35], as do the basal ruminants *Leptomeryx*, *Archaeomeryx*, and members of the Hypertragulidae [20, 26], the basal suoid *Perchoerus* and members of the Palaeochoeridae [26, 42], and members of the endemic European Cainotheriidae and Anoplotheriidae [36, 39, 43]. The extant ruminant *Tragulus napu* and the extant suid *Babyrousa babyrussa* also have a deep subarcuate fossa [26]. Furthermore, the extant camelid *Camelus dromedarius* has a shallow subarcuate fossa [26]. This character state distribution suggests that, while a shallow subarcuate fossa is shared between protoceratids and pecoran ruminants, this morphology may have evolved independently several times. Difference in subarcuate fossa depth have yet to be quantified for artiodactyls so comparisons are currently based on subjective definitions.

Perhaps a more compelling argument for a close relationship between protoceratids and ruminants—or the lack of a close relationship between protoceratids and camelids—is the absence of a mastoid fossa in protoceratids. The mastoid fossa is an indentation in the subarcuate fossa that houses the lobulus petrosus of the cerebellum [37]. Within Artiodactyla, it is only known from camelids [37], the homacodontid *Bunomeryx* [35], the endemic European artiodactyls *Cainotherium*, *Anoplotherium*, *Dichobune*, and *Xiphodon* [36, 38, 43, 44], and potentially the oreodonts *Merycoidodon* and *Ticholeptus* [35, 37]. Like the shallow subarcuate fossa, the lack of a mastoid fossa in protoceratids has been used to suggest that protoceratids are more closely allied with ruminants than with camelids [7, 25].

There are a few differences between *P. celer* and other protoceratids. *Leptotragulus* has a rostral tympanic process, a thick rim of bone bordering the ventrolateral pars cochlearis below and behind the promontorium [7]. The size of this process may have caused the *Leptotragulus* fenestra cochleae to be ventrally oriented [7]. A similarly enlarged rostral tympanic process and ventrally-oriented fenestra cochleae are present on the basal ruminants *Hypertragulus*, *Archaeomeryx*, and *Leptomeryx* [20]. No such enlarged rostral tympanic process is found on *P. celer* or the more derived protoceratid *Syndyoceras* [25]. However, the fenestra cochleae of *P. celer* opens ventrally like that of *Leptotragulus*. This suggests that an enlarged rostral tympanic process may be the ancestral condition for protoceratids, and that the ventral orientation of the fenestra cochleae was retained for some time after the rostral tympanic process was reduced.

*Protoceras celer* has a tegmen tympani fossa, which is a rostrally-directed depression on the tegmen tympani that opens towards the cerebral cavity [34]. The early artiodactyls *Diacodexis*, *Dichobune*, and *Homacodon* also have this condition [34]. Orliac and O'Leary suggested that the tegmen tympani fossa received part of the temporal lobe of the cerebrum and the trigeminal ganglion for the trigeminal nerve [45]. A tegmen tympani fossa has not been explicitly documented in other protoceratids, but Joeckel and Stavas described a well-developed shelf-like process at the rostromedial border of the *Syndyoceras* petrosal [25]. This process forms the dorsolateral border of an alisphenoid groove that may have transmitted the trigeminal nerve or ganglion [25]. *Protoceras celer* lacks such a process and does not have any structures that roof the alisphenoid in the manner depicted in CT scan of *Syndyoceras* [25]. Joeckel and Stavas suggested that *Syndyoceras* was displaying a basal artiodactyl condition because neither

camelids nor ruminants are known to have a similar shelf-like rostral process [25]. Given that the process is not present in more basal protoceratids such as *P. celer*, it is more likely that this shelf-like process is a derived condition. The morphology of *P. celer* may be the precursor to the more elaborate morphology of *Syndyoceras*—if the latter has a tegmen tympani fossa (which cannot currently be determined), the fossa may be expanded rostrally and medially to border the alisphenoid canal. This would be in line with the suppositions of previous researchers that both structures are in close association with the trigeminal ganglion [25, 45].

Protoceras celer differs from both *Leptotragulus* and *Syndyoceras* in possessing a petromastoid canal [7, 25]. This canal transmits the subarcuate artery [46], and the path of the canal can be clearly followed in the high-resolution CT scan of AMNH-VP 53523. The presence of a petromastoid canal has evolved several times in artiodactyls; it is present in extant hippopotamids, some suoids, and *C. dromedarius* [26], as well as several dichobunoids [34], several extinct suoids [42], the oreodont *Merycoidodon* [26], and the anoplotheriid *Diplobune* [39]. A petromastoid canal is also found in the mesonychid *Dissacus* [26]. Orliac and O'Leary suggested that the widespread presence of the petromastoid canal in early artiodactyls may indicate that it is an artiodactyl plesiomorphy [45]. If so, then *P. celer* has either retained or independently re-evolved a primitive condition that has been lost in other protoceratids.

## Bony labyrinth

To our knowledge, this is the first published description of a protoceratid bony labyrinth. The bony labyrinth morphology of other purported tylopods is not well-known; morphologies have only been described from cainotheriid *Cainotherium* [36, 43], the anoplotheriid *Diplobune* [39], and the oreodont *Bathygenys* [31]. However, there have been extensive descriptions of extinct and extant ruminant bony labyrinths [41, 47–50], and the bony labyrinths of the early artiodactyl *Diacodexis ilicis* and the extant suid *Sus scrofa* have also been described [31, 51].

The cochlea of *P. celer* has 2.75 turns, which is more turns than *Diplobune*, moschids, cervids, and bovids, but fewer turns than *Cainotherium* and *S. scrofa* [39, 41, 43, 47]. It is most comparable to the tragulids; most tragulids have 3.0 turns or more, but *Moschiola meminna* can range from 2.75 to 3.25 turns [48, 49, 51]. Cochlear coiling within a species often varies by 0.5 turns [49]. Using this range, the cochlea of *P. celer* is comparable to most artiodactyls, excluding *D. ilicis*, *Bathygenys*, and *S. scrofa*.

The *P. celer* cochlea has an aspect ratio of 0.80. Anything above 0.55 is considered to be a high aspect ratio, generally associated with "sharp-pointed" cochleae [32]. The aspect ratio of *P. celer* is higher than that of other artiodactyls; the highest aspect ratio previously reported is from a juvenile specimen of the tragulid *Hyemoschus aquaticus* (aspect ratio: 0.75), which also has 2.75 cochlear turns [49]. Aspect ratios can vary within a species; other juvenile specimens of *H. aquaticus* have aspect ratios as low as 0.62, and adult *H. aquaticus* specimens have aspect ratios ranging from 0.57–0.62 [49]. A high aspect ratio is derived for artiodactyls, with basal forms having ratios under 0.55 [31, 51]. The high aspect ratio of *P. celer* is likely the result of a tightly coiled basal turn rather than a high number of coils.

The vestibule of *P. celer* is typical of artiodactyls. Most taxa have a slightly inflated saccule and utricle with a clear distinction between the two structures [e.g., 39, 48, 51], although this is not the case of *Bathygenys* [31]. The vestibular aqueduct appears to originate from the common crus, but the medial end of the aqueduct could not be identified in *P. celer*. Artiodactyls generally have a vestibular aqueduct that originates either at the base of the common crus or just anterior to the common crus [e.g., 39, 48, 51], so the position of the *P. celer* vestibular aqueduct is as expected. Not much can be said about the morphology of the semicircular canals given that only one canal is preserved in AMMH-VP 53523.

## Ectotympanic

The *P. celer* bulla is located between the squamosal, basioccipital, and paroccipital process of the exoccipital. This is typical of all protoceratids [2, 25]. Joeckel and Stavas [25] observed that *Syndyoceras* has a thin bony process extending from the basioccipital to the bulla. No such process is found in *P. celer*, but this may be because of regional breakage. Scott [10] reported that the bulla and basioccipital of *Protoceras* are too closely appressed for the petrosal to be visible through the gap. There is a gap in AMNH-VP 1229, but the gap is filled with matrix and no internal structures can be observed. Scott noted that one *Protoceras* specimen had an enlarged gap because of basicranial distortion [10]. This may be the case for AMNH-VP 1229 as the specimen is dorsoventrally compressed.

The auditory bulla of *P. celer* is small and uninflated, a condition shared with all protoceratids [2, 10, 18, 25]. Poor preservation of the bulla means that its internal structure cannot be determined, but previous authors have reported that *Protoceras* joins other protoceratids in having a hollow bulla [25]. Most ruminants (except tragulids) also have a hollow bulla, whereas camelids, cainotheriids, suiforms, and some merycoidodontids have a bulla filled with cancellous bone [25, 36, 37]. Like *Paratoceras* and *Syndyoceras*, the styliform process of *P. celer* is wide and blunt [2, 25]. Other artiodactyls with small- or medium-sized bullae typically have a more slender styliform process [37].

The lateral ectotympanic contributes to the rostral portion of the external auditory meatus and the squamosal contributes to the dorsal and caudal portions. This construction is found in all protoceratids, as well as pecorans and the homacodontid *Bunomeryx* [2, 7, 25, 35]. Conversely, the external auditory meatus of camelids is primarily formed by the ectotympanic, having only a slight dorsal contribution from the squamosal [37, 52]. In cainotheriids, which are also purported tylopods, the squamosal does not contribute to the external auditory meatus at all [36].

The *P. celer* ectotympanic also extends as a ventral projection below the external auditory meatus. A similar ventral projection is present in *Syndyoceras* [25]. In both cases, the projection is filled with cancellous bone. Joeckel and Stavas posited that this projection might be homologous to the much larger "lateral plate" of the camelid bulla [25], but concluded that it could easily be an independent derivation as several artiodactyls have a similar structure [53]. The ventral projection of *P. celer* does not help to resolve this question of homology, but it does suggest that a cancellous ventral projection is common in protoceratids.

## Squamosal

Squamosal morphology is fairly conserved in protoceratids. Like others in the family, *P. celer* lacks a preglenoid process, has a slightly convex glenoid fossa, and has a low postglenoid process. A sinus venosus temporalis is present in both basal and derived protoceratids, and in several other artiodactyls including the oreodont *Merycoidodon culbertsoni* [37], the cainotheriid *Cainotherium* [36], and the camelids *Poebrotherium* and *Lama glama* [25, 37]. The sinus venosus temporalis of the basal protoceratid *Leptotragulus* is reportedly larger than that of the derived protoceratid *Syndyoceras* and of non-protoceratids [7]. The sinus venosus temporalis of *P. celer* appears to be slightly larger than that of *Syndyoceras*, but distortion of the skull makes such comparisons difficult. It does not appear to be as large as the sinus venosus temporalis of *Leptotragulus*.

A supraglenoid foramen, similar to that of the protoceratid *Paratoceras*, is present in *P. celer* [2]. To our knowledge, these are the only protoceratid taxa for which a supraglenoid foramen has been reported. The lack of its identification in previous descriptions of *Protoceras* [10, 18] suggests that the foramen may be variably present within the taxon. A supraglenoid

foramen could not be identified on AMNH-VP 53523 even though sections of the internal canal leading to the foramen are present. This may be because of poor exterior preservation or may be a true absence. We have been unable to examine additional specimens and thus cannot comment on the general distribution of the supraglenoid foramen among protoceratids.

A foramen jugular spurium was reported in one specimen of *Leptotragulus* [7] but this foramen could not be located on the *P. celer* specimens.

External exposure of the petrosal (the mastoid condition) is common in selenodont artiodactyls, although the position and amount of exposure varies among taxa [37, 53]. Typically, the mastoid sits between the squamosal dorsolaterally, the exoccipital ventrally, and the supraoccipital medially. The mastoid exposure of *P. celer* is normal in this regard, and is similar to that of other protoceratids in being a laterally-oriented thin band of exposed bone [7, 25]. Both *P. celer* and *Syndyoceras* have the typical mastoid position [25]. Norris stated that the mastoid region of *Leptotragulus* lies between the squamosal and supraoccipital, but the paroccipital processes were missing from the specimens he examined [7]. It is unclear whether there would have been mastoid-exoccipital contact if the paroccipital processes were intact. Mastoid contact has not been described for other basal protoceratids, but based on an illustration of *Leptoreodon marshi*, the mastoid does contact the exoccipital [12]. Norris described the presence of a mastoid foramen on the dorsal border of the exposed mastoid region [7]. The high-resolution CT scan of AMNH-VP 53523 does not extend far enough caudally to determine if a mastoid foramen is present, and we do not know of any published descriptions of *Protoceras* having a mastoid foramen.

## Exoccipital

The exoccipital of *P. celer* is like that of other protoceratine protoceratids [2]. *Syndyoceras* has a tight articulation between the paroccipital processes and the auditory bulla [25]. *Protoceras celer* also has a close contact between the structures, but we cannot comment on whether there is fusion because the bullar portion of the ectotympanic is missing in AMNH-VP 53523 and the CT scan of AMNH-VP 1229 is not of high enough resolution.

## Basisphenoid

*Syndyoceras* has a ventral midline groove running along the basioccipital onto the basisphenoid [25]. There is a midline groove present on the basioccipital of AMNH-VP 53523, but we cannot determine whether it continues onto the basisphenoid because the point of contact between the two bones in indistinct. A pair of ventral grooves bordering the basisphenoid midline, just rostral to the termination of the original midline groove, was figured for *Syndyoceras* [25]. These grooves are present on AMNH-VP 53523.

## Basioccipital

The basioccipitals of *Protoceras* and *Syndyoceras* have been reported to be similar in shape and structure [25]. We concur with this assessment, although we do note some additional features. Both AMNH-VP 1229 and AMNH-VP 53523 have separate hypoglossal and condylar foramina on the left side of the skull. Separate foramina are not uncommon, and this separation often occurs on only one side of the skull. Such variation is present on specimens of *Ovis* and *Lama* (pers. obvs.) and have also been documented on the mesonychid *Dissacus* [54].

*Syndyoceras* has a pronounced basicapsular groove on the dorsolateral surface of the basioccipital (Fig 7E) [25]. This groove likely carried the inferior petrosal venous sinus. *Protoceras celer* also has a basicapsular groove, but it is less pronounced. There is a faint complementary groove on the ventral surface of the petrosal, suggesting that the inferior petrosal venous sinus

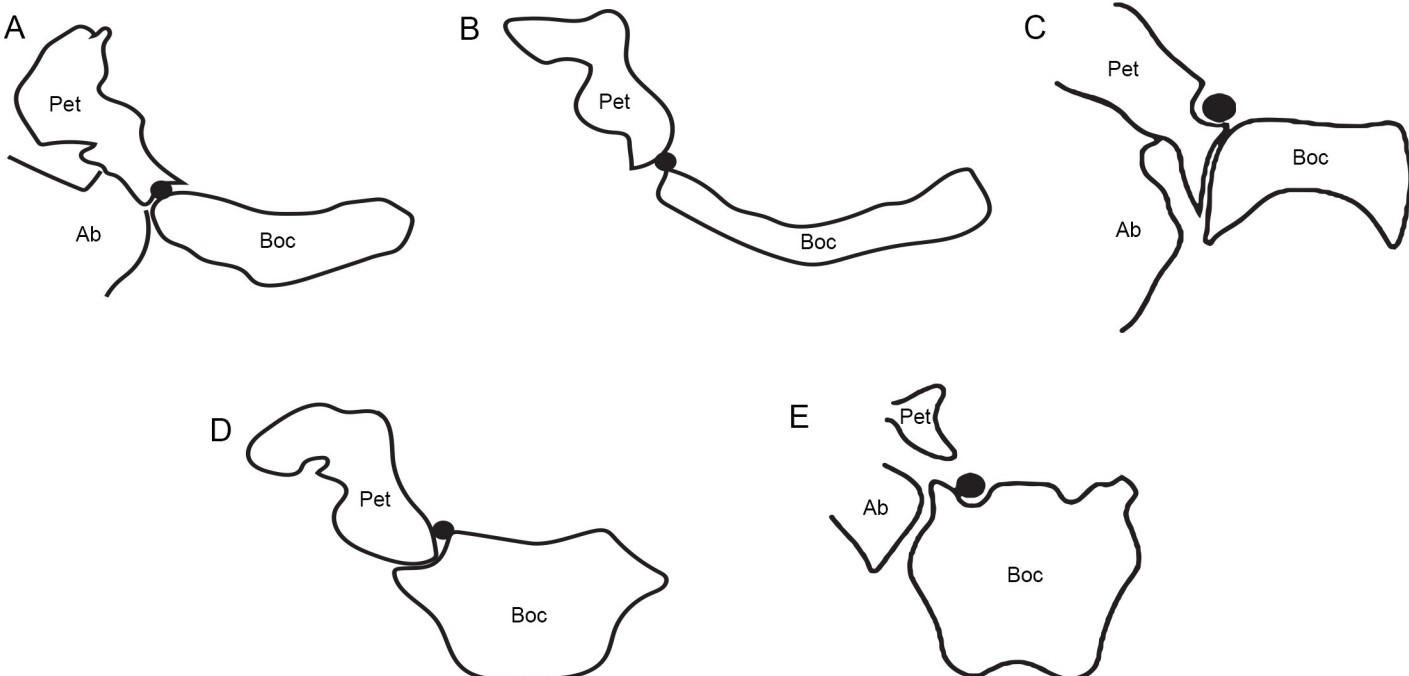

**Fig 7. Diagrammatic basicranial cross-sections showing the inferior petrosal sinus, inferred based on basicapsular groove position, in various artiodactyl families.** (A) *Lama pacos* (ZM 16018), a camelid. (B) An unidentified ruminant. (C) *Cainotherium commune* (YPM 25037), a cainotheriid. (D) *Protoceras celer* (AMNH 53523), a relatively underived protoceratine protoceratid. The bullar portion of the ectotympanic is absent in this specimen. (E) *Syndyoceras cooki* (USNM 1153), a relatively derived synthetoceratine protoceratid. The CT slice depicted here is relatively rostral compared to the other taxa; the basicapsular groove does not appear to extend farther caudally [25]. The black circle represents the inferred position of the inferior petrosal sinus. Abbreviations: Ab, auditory bulla; Boc, basioccipital; Pet, petrosal. C is after Theodor [36], E is after Norris [35].

was cradled between the two bones rather than located solely on the basioccipital (Fig 7D). *Protoceras celer* may be displaying an intermediate condition; Norris described a similar groove on the ventromedial surface of the *Leptotragulus* petrosal, but there was no discussion as to whether an accompanying basioccipital groove was present [7]. *Syndyoceras* has a small, paired sinus in the dorsal basioccipital, adjacent to the auditory bulla and immediately posterior to the basicapsular grooves. No such sinuses are present in *P celer*. Joeckel and Stavas suggested that this paired sinus was the caudal portion of the inferior petrosal venous sinus [25]. If so, the absence of this sinus in *P. celer* further indicates the minor association between the inferior petrosal venous sinus and the basioccipital.

The difference in basicapsular groove location between *Syndyoceras* and *P. celer* potentially has phylogenetic significance. Most extant artiodactyls have an inferior petrosal venous sinus that passes through the space between the auditory bulla and basioccipital [35]. Conversely, camelids, *Merycoidodon*, and *Bunomeryx* have an inferior petrosal venous sinus that is sandwiched between the basioccipital and the petrosal, much like the sinus of *P. celer* [35, 37] (Fig 7). The petrosal-basioccipital location of the sinus has been previously proposed as a tylopod synapomorphy [35]. *Cainotherium* and *Syndyoceras* appear to be the extremes of this condition; *Cainotherium* carried the inferior petrosal venous sinus entirely on the petrosal, and *Syndyoceras* carried the inferior petrosal venous sinus entirely on the basioccipital (Fig 7) [25, 36]. The confinement of the inferior petrosal venous sinus to the basioccipital has been used as evidence against a tylopodan affiliation for *Syndyoceras* and protoceratids as a whole [25]. The discovery that *P. celer*, a protoceratid basal to *Syndyoceras*, has a petrosal-basioccipital location for the sinus brings this conclusion into question. However, such a position does not

necessitate that protoceratids are tylopods. Camelids and *Bunomeryx* both have a prominent ventromedially directed "flange" on the petrosal that roofs the basicapsular groove [25, 35]. *Leptotragulus* and *P. celer* lack such a flange; the ventral border of the petrosal is rounded in both taxa [7]. This suggests that the petrosal-basioccipital condition observed in *P. celer* may be independently derived. The small size and short length of the basicapsular groove on the basioccipital could indicate that the inferior petrosal venous sinus was in the process of migrating from an unknown ancestral condition to the derived condition of *Syndyoceras* (Fig 7). Several extant ruminants, all lacking a ventromedial flange, have a basicapsular groove on the petrosal [26], so the presence of such a groove on *Leptotragulus* is not particularly informative. The endocranial morphology of more basal protoceratids will need to be examined to determine what the ancestral protoceratid condition may be.

## Body masses and agility scores of *P. celer*

Body masses were calculated for both AMNH-VP 1229 and AMNH-VP 53523, using total skull length and basicranial length in both the "all ungulates" and "ruminants only" regressions of Janis [29]. Body mass estimates based on total skull length were consistently larger than body mass estimates based on basicranial length. When total skull length was used, the body mass estimates of AMNH-VP 1229 were larger than those of AMNH-VP 53523. This is likely because the anterior tip of the AMNH-VP 53523 rostrum is broken off, leading to the total skull length being underestimated. When basicranial length was used, AMNH-VP 53523 was estimated to have a much larger body mass than AMNH-VP 1229. This is in line with previous observations that *P. celer* exhibits size-based sexual dimorphism [2].

The body mass estimates of AMNH-VP 1229, which has a complete rostrum, are quite different depending on whether the total skull length or the basicranial length are used. We are unsure as to the cause of this difference, but we suggest that the discrepancy may be a result of measurement methods. Janis [29] calculated total skull length by combining the posterior skull length, lower molar row length, and anterior jaw length, whereas we calculated the total skull length of AMNH-VP 1229 by taking a single measurement from the tip of the rostrum to the occiput. We cannot use the method proposed by Janis [29] because we cannot measure the lower molar row of AMNH-VP 1229. Therefore, we cannot determine if measurement methods are truly the cause of the discrepancy.

The completeness of the AMNH-VP 53523 left anterior semicircular canal allowed us to estimate an agility score for *P. celer*. The estimated scores, based on two body mass predictions, were 3.29 and 3.00. Agility scores are integer values that can range from 1 to 6, with 1 corresponding to the least agile mammals (e.g., sloth) and 6 corresponding to the most agile mammals (e.g., squirrel) [30]. The cursorial artiodactyl *Gazella bennetti* has an agility score of 3.37 while the slower moving artiodactyl *S. scrofa* has an agility score of 2.53 [33]. An intermediate artiodactyl, *Camelus dromedarius*, has an agility score of 2.67 [33]. These values are derived from a predictive equation that incorporates all three semicircular canals. When only the anterior semicircular canal is used to calculate agility scores, as was necessitated for *P. celer*, *G. bennetti* has a score of 3.29, *C. dromedarius* has a score of 2.73, and *S. scrofa* has a score of 1.85; the scores have a slightly larger range but are still comparable [33]. Based on these data, the agility scores of *P. celer* suggest that it was an intermediate to cursorial animal, an interpretation that is supported by its postcranial morphology.

## The identity of AMNH-VP 645

In her monograph on artiodactyl petrosals, O'Leary [26] described and figured a petrosal, AMNH-VP 645, referred to *P. celer*. The skull of AMNH-VP 645 was previously assigned to *P.*

*celer* [2], but we cannot determine whether the AMNH-VP 645 petrosal belongs to the same individual; to our knowledge, there is no record of the petrosal being collected in association with the skull or being dissected out of the skull after collection. The morphology of the AMNH-VP 645 petrosal contrasts with the morphology described for basal (*Leptotragulus*) and highly derived (*Syndyoceras*) protoceratids, implying reversals in the interpretation of several characters such as the presence of a deep subarcuate fossa. Our description of an *in-situ* petrosal of *P. celer* (AMNH-VP 53523) is in line with the morphology of other protoceratids and contrasts with the morphology of AMNH-VP 645 Unlike AMNH-VP 645, AMNH-VP 53532 has a shallow subarcuate fossa, lacks a distinct notch on the tympanic face dorsal to the epitympanic wing, and has a large, ventrolaterally directed mastoid process. These differences may be the result of polymorphism but, to our knowledge, artiodactyl petrosals have not been documented to exhibit this level of intraspecific variation. Compared to the AMNH-VP 53523 petrosal, the AMNH-VP 645 petrosal is generally more rounded and lacks a large mastoid process. This morphology could suggest that the AMNH-VP 645 petrosal comes from an immature individual [55]. However, the presence of a deep subarcuate fossa renders this possibility unlikely [40, 55, 56]. Given that the identity of AMNH-VP 53523 is unquestionably *P. celer*, we suggest that the AMNH-VP 645 petrosal is either an incredibly aberrant specimen, or, more likely, was assigned to *P. celer* in error. A re-examination of the specimen could provide clarification.

## Conclusion

Basicranial morphology, particularly petrosal morphology, has repeatedly been used as evidence for a close relationship between protoceratids and ruminants. These characters include the presence of an endocranial ridge, the lack of a ventromedial flange, the shallow subarcuate fossa, and the lack of a mastoid fossa. However, none of these features are unique to protoceratids and ruminants. The basicranial morphology of *P. celer*, a phylogenetically intermediate protoceratid, is similar to both basal (*Leptotragulus*) and derived (*Syndyoceras*) forms, suggesting that basicranial morphology is conserved in the family. *Protoceras celer* exhibits some intermediate conditions which align with the hypothesized phylogenetic position of the taxon; the basicrania of *P. celer* may document a transition in the orientation of the fenestra cochleae and the position of the basicapsular groove. *Protoceras celer* also possesses a petromastoid canal, which is an as-yet undocumented structure in protoceratids. The petromastoid canal is highly homoplastic in artiodactyls so the presence of such a structure in *P. celer* is not wholly surprising. The basicranial morphology of *P. celer* does not greatly illuminate the evolutionary relationships between protoceratids and other selenodont artiodactyls; however, the morphology of *P. celer* indicates that protoceratid basicrania did not undergo drastic changes during their evolution, despite derived members of the family acquiring extreme morphologies in other regions of the skull.

## Acknowledgments

We thank C. Norris and J. Galkin at the American Museum of Natural History, New York, and W. Fitch at the University of Calgary for access to specimens, and A. Mellone at the American Museum of Natural History, New York for consultation and images of AMNH-VP 645. We thank M. Colbert and A. Mote at the High-Resolution C-Ray CT Facility, University of Texas at Austin, for scanning and initial image processing of AMNH-VP 1229 and AMNH-VP 53523, and G. McRae, I. Pauchard, Y. Zhu, J. Allan, and A. Cooke at the Centre for Mobility and Joint Health, University of Calgary, for scanning UCMZ 1989.47 and UCMZ 1975.496. We acknowledge that the specimens we used in this study were collected from the ancestral lands of the Oglala Lakota people.

## Author Contributions

**Conceptualization:** Jessica M. Theodor.

**Formal analysis:** Selina Viktor Robson.

**Funding acquisition:** Jessica M. Theodor.

**Investigation:** Selina Viktor Robson, Brendon Seale.

**Project administration:** Jessica M. Theodor.

**Resources:** Jessica M. Theodor.

**Software:** Jessica M. Theodor.

**Supervision:** Jessica M. Theodor.

**Visualization:** Selina Viktor Robson, Brendon Seale, Jessica M. Theodor.

**Writing – original draft:** Selina Viktor Robson.

**Writing – review & editing:** Brendon Seale, Jessica M. Theodor.

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
