## [Decision Letter · Decision Letter 0]

26 May 2021

PONE-D-21-13984

The petrosal and basicranial morphology of Protoceras celer

PLOS ONE

Dear Dr. Robson,

Thank you for submitting your manuscript to PLOS ONE. After careful consideration, we feel that it has merit but does not fully meet PLOS ONE’s publication criteria as it currently stands. Therefore, we invite you to submit a revised version of the manuscript that addresses the points raised during the review process.

All three reviewers recommended publication of your article following minor revisions. These are largely of a structural nature, or else refer to minor points of clarification or discussion. The only substantive concern for this study is related to the availability of the CT scans - no mention is made, but ideally these should be made available in some online and open repository.

We look forward to receiving your revised manuscript.

Kind regards,

Julien Louys

Academic Editor

PLOS ONE

Journal Requirements:

2. We noted in your submission details that a portion of your manuscript may have been presented or published elsewhere. [Preliminary results were published as a conference abstract for the Society of Vertebrate Paleontology meeting in 2011 (p. 203). The published results are a minor component of the current paper, and some of the results presented at the conference have been updated upon further analysis.

Related Manuscript file type was not available in the dropdown options. The conference abstract has been uploaded as an Other file type with the description of "Related Manuscript".]

Please clarify whether this conference proceeding or publication was peer-reviewed and formally published. If this work was previously peer-reviewed and published, in the cover letter please provide the reason that this work does not constitute dual publication and should be included in the current manuscript.

Reviewers' comments:

Reviewer's Responses to Questions

**Comments to the Author**

1. Is the manuscript technically sound, and do the data support the conclusions?

Reviewer #1: Yes

Reviewer #2: Yes

Reviewer #3: Yes

2. Has the statistical analysis been performed appropriately and rigorously? 

Reviewer #1: N/A

Reviewer #2: Yes

Reviewer #3: N/A

3. Have the authors made all data underlying the findings in their manuscript fully available?

Reviewer #1: No

Reviewer #2: No

Reviewer #3: Yes

4. Is the manuscript presented in an intelligible fashion and written in standard English?

Reviewer #1: Yes

Reviewer #2: Yes

Reviewer #3: Yes

5. Review Comments to the Author

Reviewer #1: The descriptions are sharp and to the point, concise and very easy to follow, that of the petrosal is particularly accurate. Conclusions are sound and cautious (I really appreciate this), and this is a nice contribution to understand the evolutionary history of the ear region in artiodactyls. I only have very minors comments listed below. I also have made some comments on an annotated pdf.

Organization: I found it a little bit surprising to have the description of the petrosal in the middle of the rest of the cranium and to find the description of the exoccipital after the petrosal. Why not separating the description of the petrosal from that of the rest of the cranium as there is a focus on the petrosal? Also I found it weird to have the bony labyrinth before the description of the petrosal.. I would have included the agility scores within the bony labyrinth paragraph, or just after it.

What would you think of the following organization: general description of the cranium, then auditory region with i) petrosal, ii) bony labyrinth, iii) auditory bulla / auditory meatus, iv) other bones surrounding the petrosal? This is only a suggestion.

Data accessibility: There is no mention about accessibility of the 3D models, or CT scan data. 3D models should be accessible online (e.g., MorphoSource, MorphoMuseuM), or upon request, but this has to be stipulated somewhere in the text (sorry if I missed it).

Figures:

Figure 1 – I don’t really see the point to add labels on that figure, it is way too small to properly see the structures, especially the bulla. It might be nice to add some focus to specific regions of the cranium that are highlighted in the text, or just remove the labels.

Figure 2 – an interpretation drawing might be useful here as the ventral view is not very easy to read because of the 3D rendering. Especially the shape of the bulla is very difficult to decipher, which is a little bit problematic as it is described in the text.

Maeva Orliac

Reviewer #2: Dear Authors, dear editor,

I have reviewed the article entitled “The petrosal and basicranial morphology of Protoceras celer”. Protoceratidae are an interesting and enigmatic family of selenodont artiodactyls. This clade has been considered as part of the basal Tylopoda or the derived Ruminantia. The description here encompass the outer and inner morphology of the basicranial region, including the bony labyrinth and the dorsal surface of the petrosa. Unfortunately, some structures cannot be observed due to preservation and maybe some scanning parameters. This is a precise and complete work. The manuscript is well-written. It would be nice to have access to the 3D models. Maybe it would be even more easier to follow the discussion if the discussion was in 2 parts : The basicrania in artiodactyls, the protoceratidae basicrania.

I just have some minor comments.

Line 61: Archaeomeryx, which is considered as the basalmost ruminant, do have upper incisors (Janis & Scott 1987, p9)

Line 64: indeed this is a synapomorphy but not an autapomorphy since this is also observed in other clades as the primitive artiodactyls Amphymericidae (Erfurt & Métais 2007).

Line 68-70 do you have a reference?

Line 79-82 What about :

At the turn of the twenty-first century, novel information became available. The endocranial morphology of the basal “leptotraguline” protoceratid Leptotragulus was described from physical dissections of the fossil [1] and the derived synthetocerine protoceratid Syndyoceras were described was described from computed tomography (CT) scans [25].

Line 83 Joeckel and Stavas [25] and Norris [1].

Norris is the first reference not the 7

Line 111 Could you precise the age of the Poleside member? Maybe with million of year and early Oligocene (?)

Line 126 0.5 mm thickness. Do you mean that the voxel size is 500�m?

Line 133 Do you mean 74.36��m of voxel size? This may explain the resolution of the semicircular canals. In Mennecart and Costeur 2016 (Tragulidae paper) they used 40 �m.

Line 138-140 which species and which scanning resolution?

Line 145 I think you mean Spoor 29 and not Janis 28.

Line 148 I think calculating the BM for both could be done easily and would be interesting for data on sexual dimorphism.

Figure 1. It would be nice to have it duplicated with the outline of the bones

(see this example from PlosOne https://journals.plos.org/plosone/article?id=10.1371/journal.pone.0185679)

Line 177 “on the nasal” not the maxillary? If it is the nasal, then the nasal bones are long (cf line 175). An noted with name of the bones and outlines would help in this case.

Figure 2. It would be nice to have the outline of the bones

Figure 3 is it 70 or 500��m?

Line 240 It would be nice to have the 3D model to help to follow the description

Figure 4 I would recommend to clean the 3D model deleting (or coloring) the acoustic nerve, the greater petrosal nerve, the petromastoid canal. The cochlea could be “smoothed”/completed using the 3 different views of reconstruction.

Line 294 please delete P. celer

Line 296 What is the pars canicularis?

Line 339 “The cochlear aqueduct housed the perilymphatic duct in life.” This is not description and do you have a reference?

Line 372 Janis proposed 40 Kg. Could you please calculate the weight for the second specimen to have an idea of the sexual dimorphism?

Line 374-377

When applied to the appropriate agility predictive equation (see Materials and Methods), we recover two agility scores. Using the full skull length body mass, we predict an agility score of 2.97. Using the basicranial body mass, we predict an agility score of 3.057.

Is it what you mean?

The agility score obtained from the body mass calculated from the skull length is of 2.97 while it is of 3.057 using the body mass based on the basicranial.

Line 390 the Protoceratidae Paratoceras

Line 417-418 Joeckel and Stavas [25] observed that Syndyoceras has a thin bony process extending from the basioccipital to the bulla.

Line 419-421 Scott [10] reported that the bulla and basioccipital of Protoceras are too closely appressed for the petrosal to be visible through the gap.

Line 449 Could you specify the family of all these artiodactyls.

In this article Cainotheriidae and Anaplotheriidae are thought to be sister taxa of the ruminant

https://www.researchgate.net/publication/346096394_A_new_Cainotherioidea_Mammalia_Artiodactyla_from_Palembert_Quercy_SW_France_Phylogenetic_relationships_and_evolutionary_history_of_the_dental_pattern_of_Cainotheriidae

Line 478 which Camelid?

The Homacodontidae/basal artiodactyl Bunomeryx

What about the Merycoidodontidae (Tylopoda?) ? There are closer chronologically with the Protoceras than living ruminants and maybe more closely related. There is a Merycoidodon described in O’Leary (Fig 74)

Line 495 the Eocene camelid Poebro and the extant Lama

Line 506 Indeed + there is a lack of quantification of the structure so the definition of deep or shallow may vary.

Line 512 Is there a mastoid fossa in Merycoidodon?

Line 538 likely that this

Line 588 the basicapsular groove is variable within the ruminants

Line 589 what means an intermediate protoceratine?

Line 616-917 In ruminants the basicapsular groove is variable and seems to be phylogenetically informative. It has been used in cladistics analyses published in PlosOne (Aiglstorfer et al. 2017, Mennecart et al. 2021)

https://journals.plos.org/plosone/article?id=10.1371/journal.pone.0185679

https://journals.plos.org/plosone/article?id=10.1371/journal.pone.0244661

Line 645-647 Please consider that the external morphology of the petrosal suffers of a strong ontogenetic shape allometry. The specimen lacks a mastoid region and is very rounded as observed in calf (Costeur et al. 2017). Moreover, since the petrosal is isolated, it can be a juvenile.

https://onlinelibrary.wiley.com/doi/full/10.1111/joa.12549

However the subarcuate fossa is extremely deep and I fully agree with your conclusion.

Line 657 there should not be references in the conclusion

Line 719 Archaeomeryx should be in italic

The doi website is strange

Line 740 The doi website is strange

Best regards,

Bastien Mennecart

Reviewer #3: This paper is an important contribution to the basicranial and petrosal morphology of protoceratids. As you outline, the phylogenetic position of this group is controversial, and the new data you describe help resolve this issue. Your study is also the first to describe the inner ear of a protoceratid, and thus adds to a growing database of cetartiodactyl inner ears.

My comments are fairly minor. My most substantive comments involve your interpretation of the differences you observe between the specimens you CT scanned and the separate petrosal described by O'Leary (2010). Your preferred hypothesis is that AMNH 645 is misidentified and more likely to be that of Poebrotherium. However, O'Leary 2010 also described and figured the petrosal of Poebrotherium, and they do not look similar. Instead AMNH 645 looks a lot like the Protoceras petrosal you describe. In fact the endocranial views (your fig. 5C and fig. 41 of O'Leary 2010) are nearly a perfect match. They both have very similar proportions and a small, pointed epitympanic wing. While the polarity of many features are unknown, this leads to me think that AMNH 645 is correctly identified and that either 1) the mastoid fossa is polymorphic in this species or 2) that the CT did not detect the fossa because of matrix that was similar in density to bone. Other comments are listed below.

1) Line 86 - this is an odd way to refer to a study, I suggest using the author's name (O'Leary) instead of such a passive form.

2) Line 90 - In fig. 41 of O'Leary 2010 there is clearly an endocranial ridge. The text of that paper say otherwise but this is probably an error.

3) Body size estimates - I suggest showing values for the "all artiodactyl" estimate since the position of protoceratids is unclear. If the estimate is off please explain why and show that.

4) Morphological description - Since the morphology of this taxon has been described, I would suggest removing those parts that are redundant. I would add a few lines indicating what features support the referral of the specimens you CT scanned to Protoceras, particularly in light of the differences observed with AMNH 645.

5) Line 276 - What is the later process of the epitympanic wing?

6) Line 279 - Please label this in fig. 5. Could be mislabeled as "petromastoid flange"

7) Line 318 - What is the mastoid plate?

8) Line 320 - Please provide a citation for the "tegmen tympani fossa". This is not a commonly used term.

9) Line 359 - spelling

10) Line 549-551 - Please check. I do not think this has been described in Dissacus. Are you confusing the petromastoid and post-temporal canals?

11) Fig. 5 - please label the endocranial ridge

6. PLOS authors have the option to publish the peer review history of their article (what does this mean?). If published, this will include your full peer review and any attached files.

Reviewer #1: **Yes: **Maeva J Orliac

Reviewer #2: **Yes: **Mennecart Bastien

Reviewer #3: No

---

## [Author Response · Author response to Decision Letter 0]

17 Jun 2021

Editor

1. We have added the corresponding author’s email address to the title page, and we have changed our Level 1 and Level 2 headings to match journal requirements. We did not have any Level 3 headings to modify. To our knowledge, our figure captions and equations are formatted correctly. Equations were inserted using the Insert � Equation function in Word. We do not have any inline equations in our manuscript. All inline symbols were inserted using the Insert Symbol function. We have done our best to name the files according to the PLOS ONE style requirements. 

2. The conference abstract was not peer-reviewed and/or formally published and should not violate the rules of dual publication. 

3. We apologize for confusion about data accessibility. We were under the impression that we needed to provide the information in the Data Accessibility section but not the manuscript. We have added the DOIs and relevant instructions to the body of the manuscript. We are unable to make some data publicly accessible because the data are owned by the American Museum of Natural History (AMNH). We have provided instructions in our manuscript on whom to contact at the AMNH to request permission to download the data. Restrictions are outlined by the AMNH here: https://www.amnh.org/research/paleontology/3d-scanning, under “Copyright, Licensing, Use and Distribution of 3-D data generated from specimens in the AMNH Division of Paleontology collections”. We see no reason why the AMNH would deny a download request, and the specimens are accessible for examination and re-scanning by interested parties, but the final decision lies with the museum. All data not owned by the AMNH are publicly accessible on MorphoSource. 

4. We do not have any supporting information files for this manuscript. 

Note: We have updated the phrasing in our acknowledgements to reflect the preferences of the named parties. 

Reviewer 1

Line 99: We have changed “explanations” to “scenarios”. 

Line 101-103: We recognize that including conclusions is a spoiler. We included this statement because PLOS asks that we conclude our introduction “with a brief statement of the overall aim of the work and a comment about whether that aim was achieved”. We have attempted to revise our statement to reduce spoilers while still complying with the journal. 

Line 129: The resolution was in pixels. We have added voxel sizes and removed the resolutions because they are now redundant. 

Line 167: We have clarified that we are discussing the cranium.

Line 228-230: We have updated our figures to better-show the morphology.

Line 384: We have added some text to express the relevance of cainotheriids. The text does not fit particularly well in this line, so we added it to the next mention of cainotheriids. 

Line 496-497: Good point about the character polarities. We have modified the text and no longer suggest that a deep subarcuate fossa is unusual. 

Line 511-512: We agree that the mastoid fossa could easily be a symplesiomorphy. So far, most descriptions of basal artiodactyl petrosals have not made mention of a mastoid fossa, but we would be interested in exploring this further. It is possible that a mastoid fossa was present and was simply not described. Given that we currently have limited data, we are choosing not to make such an assumption in our manuscript and are rather listing the taxa that are known to have a mastoid fossa. 

Organization: We have reorganized the descriptions as per the reviewer’s suggestion. We thank the reviewer for providing us with an outline for the organization. We have chosen to keep the body mass and agility score estimates separate because, while they are inferences based on the data in our description, they are not actually part of the description. 

Data accessibility: We were unsure of where to put the DOIs, so we reported them in the submission form but not the manuscript. We are in contact with the editor to discuss data availability. We apologize for not including the relevant information in the manuscript. 

Figure 1: We have modified this figure by adding outlines to the specimens. We think that this enhances the visibility of the structure we discuss in the text, including the auditory bulla.

Figure 2: We have modified this figure by adding outlines to the CT renderings. As with Figure 1, we think that this enhances the visibility of structures which were previously unclear. 

Reviewer 2

Line 61: We have clarified the sentence to say most ruminants. 

Line 64: We fully agree that a cubonavicular is not a ruminant autapomorphy and we were not intending to suggest differently. The purpose of our statement was to indicate that protoceratids lack a cubonavicular and, as such, lack an important ruminant synapomorphy. We did not refer to amphimerycids as we did not think they were relevant to the discussion at hand. We have left the text as-is because we are concerned that mention of amphimerycids would detract from the comparison of ruminants and protoceratids, but we welcome suggestions on rephrasing if the reviewer feels strongly on this point. 

Line 68-70: We have added several references that discuss plesiomorphic and convergent morphology in artiodactyls. 

Line 79-82: We have changed the wording to what the reviewer suggested. We thank them for helping us with phrasing. 

Line 83: We have added the numbered references. The numbers have changed since the first submission because of manuscript reorganization.

Line 111: We have added dates for the Poleslide member. 

Line 126: We have changed the wording to indicate a voxel size of 500 microns.

Line 133: We have changed the wording to indicate a voxel size of 63.4765 x 63.765 x 0.07436 µm. We do not think that the relatively large voxel size would have much of an impact on the semicircular canal resolution as the skull of Protoceras is much larger than the skulls of the tragulids, but we will keep this in mind for future scanning endeavors. 

Line 138-140: We have added the taxon names and scanning parameters. The comparative data are uploaded to MorphoSource with public access. 

Line 145: The reviewer is correct. We did mean Spoor et al. (2007) and we have fixed the citation. 

148: We have added body mass calculations for AMNH-VP 1229 and have included a body mass comparison in our discussion. 

Figure 1: We have modified this figure by adding outlines to the specimens. We thank the reviewer for the suggestion and for pointing us to example images.

Line 177: We did mean the maxillaries, not the nasals, and we have amended our description. 

Figure 2: We have modified this figure by adding outlines to the CT renderings. 

Figure 3: All figures of AMNH 53323 are from the high-resolution scan. 

Line 240: We agree that it would be nice to provide the 3D models, but all scan data, including the 3D renderings, belong to the AMNH and therefor cannot be provided publicly. We have done our best to illustrate our descriptions using our figures.

Figure 4: We have added colours to the 3D model to help distinguish features. We were unable to enhance the quality of the cochlea as it is already smoothed and reconstructed based on all three views; unfortunately, the bony labyrinth of the specimen is simply not that well preserved. 

Line 294: P. celer has been deleted.

Line 296: The “pars canicularis” is an unfortunate misspelling of “pars canalicularis”. We have fixed the mistake. 

Line 339: We have removed this sentence as we agree it is not part of the description.

Line 372: The 40 kg proposed by Janis was based on the lower molar row length, which is not something we could calculate as we only have skulls. Our posterior skull length calculation produced a body mass of 22.0 kg and our total skull length calculation produced a body mass of 27.3 kg. Based on the same metrics, Janis estimated a body mass of 29.4 (SD 28.6) and 37.0 (SD 10.0), respectively. This is why we said that our estimates were similar to those from Janis. We have elaborated on our discussion of body mass, including a comparison to the second specimen and a discussion of potential sources of error. We thank the reviewer for encouraging us to explore the matter further. 

Line 374-377: The reviewer was correct in interpreting our meaning. We have amended our language to make ourselves clearer.

Line 390: We have clarified that Paratoceras is a protoceratid. 

Line 417-418: We have added the numeric citation in the correct location.

Line 419-421: We have added the numeric citation in the correct location.

Line 449: We have added the family names to the taxa. 

We included anoplotheriids and cainotherioids under the classification of “purported tylopods” because they have frequently been referred to Tylopoda in the literature, even if this position is disputed (e.g., we consider protoceratids to also be purported tylopods). We used the term “purported” to indicate that not all listed taxa necessarily belong to the Tylopoda. We have kept the sentence as-is for the moment as we hope this clarification will suffice, but we would be happy to rephrase if the reviewer has other suggestions. 

Line 478: We are referring to both extinct and extant camelids. We have clarified this in the sentence, and we have clarified that Bunomeryx is a homacodontid. Also, thank you for noting that we omitted Merycoidodon from this list. The taxon has now been added. 

Line 495: We have now indicated that Poebrotherium is extinct and Lama is extant. 

Line 506: Absolutely! Some quantification would certainly be welcome. We have added a sentence expressing this problem at the end of the paragraph.

Line 512: Merycoidodon has a shallow depression which may be a reduced mastoid fossa, and Ticholeptus might have a deeper mastoid fossa. We have added a comment about oreodonts to our discussion of the mastoid fossa. 

Line 538: We have fixed the duplicate word.

Line 588: This figure is a modified version of figures published by Norris (1999) and Theodor (2010). Thanks to the reviewer’s comments, we have further modified the figure to show that ruminants have a basicapsular groove. We have also changed the figure heading to indicate that the basicapsular groove is being used to infer the position of the inferior petrosal sinus. 

Line 589: By intermediate, we mean a protoceratid that is neither basal nor highly derived. We have changed the wording to say “relatively underived” rather than “intermediate”. Hopefully this clarifies our meaning for the figure. We have retained the word “intermediate” in other portions of the manuscript where contextual cues make our meaning more clear. 

Line 616-617: We used extant ruminants to illustrate our point that the presence of a basicapsular groove cannot solely be used to infer the location of the inferior petrosal sinus. We did not cite Aiglstorfer et al., (2017) and Mennecart et al. (2021) because, while excellent descriptions, these papers focused on extinct ruminants and did not infer the location of the inferior petrosal sinus. We agree that the position of the basicapsular groove may be phylogenetically important and should be explored further, but this was not the intended focus of the example.

Line 645-647: We have added additional discussion about the identity of AMNH-VP 645 following the reviewer’s advice. We thank the reviewer for these helpful insights. 

Line 657: We have removed the reference. 

Line 719: Archaeomeryx became italicized when we refreshed out citations. The DOI was for the wrong article and we have removed it. 

Line 740: The DOI appears to be correct, and we have confirmed that it links to the right article. We do not know why it previously looked odd. 

Reviewer 3

We thank the reviewer for directing our attention back to the figures in O’Leary (2010). We agree that, aside from the subarcuate fossa, the endocranial surfaces of AMNH-VP 645 and our described Protoceras petrosal are similar. We have re-checked our scans and have found no evidence of infilling, and we do not think that this is a likely scenario given that the path of petromastoid canal through the subarcuate fossa is clearly defined in our scan. We cannot discount potential polymorphism in the taxon and we have added such a suggestion to our manuscript, although we do note that subarcuate fossa polymorphism has not been described in other artiodactyl taxa. 

In addition to the discrepancy in subarcuate fossa size, we have noted a couple of additional differences between specimens. AMNH-VP 645 has a distinct notch on the tympanic face dorsal to the epitympanic wing. Our specimen lacks this notch. There is also a large difference in the shape of the mastoid region; AMNH-VP 645 lacks a distinct mastoid process whereas our described specimen has a large process. We have added these comments to the paper. One reviewer has suggested that these differences could be ontogenetic in nature—AMNH-VP 645 may be from a juvenile—but observed that the deep subarcuate fossa did not support this hypothesis. We have also added this aspect of discussion to our paper. We have removed our suggestion that AMNH-VP 645 may be from a basal camelid as we agree that this speculation is unnecessary and requires more data to be supported. 

1. Line 86: We agree our phrasing was awkward, and we have changed it to an active voice.

2. Line 90: Thank you for drawing our attention to Fig. 41. We have removed our statement about the endocranial ridge. 

3. We have added additional discussion of the body size estimates, including estimates for the female skull and comments on why some of the size estimates may be off. 

4. We acknowledge that some of our descriptions are redundant with previous descriptions of Protoceras celer. However, we have chosen to retain our descriptions given that AMNH-VP 53523 and AMNH-VP 1229 have not previously been described in detail.

We have added a paragraph explaining why these two specimens have been referred to Protoceras. We thank the reviewer for the suggestion. 

5. Line 276: This should say lateral process. We checked our manuscript and it appears correct in our version, which is why the line does not show any tracked changes. We are not sure why it was incorrect on the reviewer’s copy, but we apologize for the error. 

6. Line 279: We have relabeled the figure with “posteromedial” flange. 

7. Line 318: The mastoid plate is a structure present in extant suines. We have added a reference (O’Leary, 2010) for a definition. 

8. Line 320: We have added a reference (Orliac and O’Leary, 2014). 

9. Line 359: We have corrected the spelling error.

10. Line 549-551: We are a bit confused as to what is being asked here. The reviewer says that a structure has not been described in Dissacus, and then goes on to suggest we might be confusing the structure with the petromastoid canal. We are confused because the paragraph in question is indeed discussing the petromastoid canal and, in our statement about Dissacus, we only refer to the petromastoid canal. We are not sure what structure the reviewer meant, and we would greatly appreciate clarification so that we can correct any errors we have made. 

11. Fig. 5: We have labeled the region of the endocranial ridge. The ridge itself is best viewed in cross-section, as shown in Figure 6, but we hope that indicating the correct region is helpful.

---

## [Editor Report · Decision Letter 1]

8 Jul 2021

The petrosal and basicranial morphology of Protoceras celer

PONE-D-21-13984R1

Dear Dr. Robson,

We’re pleased to inform you that your manuscript has been judged scientifically suitable for publication and will be formally accepted for publication once it meets all outstanding technical requirements.

Kind regards,

Julien Louys

Academic Editor

PLOS ONE
---

## [Editor Report · Acceptance letter]

21 Jul 2021

PONE-D-21-13984R1 

The petrosal and basicranial morphology of *Protoceras celer*

Dear Dr. Robson:

I'm pleased to inform you that your manuscript has been deemed suitable for publication in PLOS ONE. Congratulations! Your manuscript is now with our production department. 

Kind regards, 

on behalf of

Dr. Julien Louys 

Academic Editor

PLOS ONE